# On the Inductive Bias of Stacking Towards Improving Reasoning

**Nikunj Saunshi**[*]
Google Research
nsaunshi@google.com

**Stefani Karp**
Google Research
stefanik@google.com

**Shankar Krishnan**
Google Research
skrishnan@google.com

**Sobhan Miryoosefi**
Google Research
miryoosefi@google.com

**Sashank J. Reddi**
Google Research
sashank@google.com

**Sanjiv Kumar**
Google Research
sanjivk@google.com

## Abstract

Given the increasing scale of model sizes, efficient training strategies like gradual stacking [Gong et al., 2019, Reddi et al., 2023] have garnered interest. Stacking enables efficient training by gradually growing the depth of a model in stages and using layers from a smaller model in an earlier stage to initialize the next stage. Although efficient for training, the model biases induced by such growing approaches are largely unexplored. In this work, we examine this fundamental aspect of gradual stacking, going beyond its efficiency benefits. We propose a variant of gradual stacking called MIDAS that can speed up language model training by up to 40%. Furthermore we discover an intriguing phenomenon: MIDAS is not only training-efficient but surprisingly also has an inductive bias towards improving downstream tasks, especially tasks that require reasoning abilities like reading comprehension and math problems, despite having similar or slightly worse perplexity compared to baseline training. To further analyze this inductive bias, we construct *reasoning primitives* – simple synthetic tasks that are building blocks for reasoning – and find that a model pretrained with stacking is significantly better than standard pretraining on these primitives, with and without fine-tuning. This provides stronger and more robust evidence for this inductive bias towards reasoning. These findings of training efficiency and inductive bias towards reasoning are verified at 1B, 2B and 8B parameter language models. Finally, we conjecture the underlying reason for this inductive bias by exploring the connection of stacking to looped models and provide strong supporting empirical analysis.

## 1 Introduction

With the advent of very large deep learning models, efficient training to reduce the compute and time requirements is becoming increasingly important. Along with efficient optimization procedures, there has been a surge in interest to design efficient training strategies. One practical approach is to use smaller models to initialize larger models. Usually, this results in much faster convergence compared to vanilla training [Chen et al., 2022, 2016, Gong et al., 2019, Reddi et al., 2023, Wang et al., 2023, Li et al., 2023, Kim et al., 2023, Yao et al., 2024, Wang et al., 2024]. Stacking and growing based approaches have particularly gained traction recently. For instance, gradual stacking [Reddi et al., 2023] is a prominent approach where in each stage the last few layers of the model

---

[*]Corresponding author

38th Conference on Neural Information Processing Systems (NeurIPS 2024).

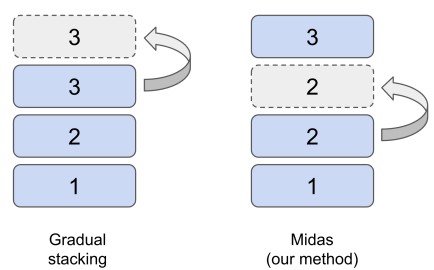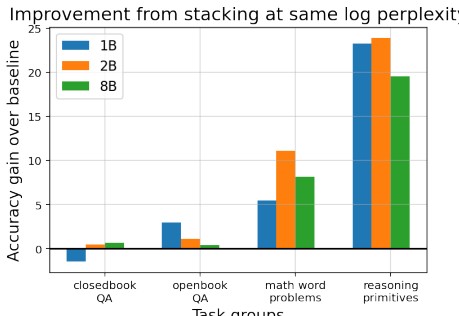

Figure 1: (a) Pictorial depiction of gradual stacking and MIDAS. (b) Accuracy improvements (in %) for model trained with MIDAS over baseline for various task groups, despite having the same perplexity. For both 1B, 2B and 8B models, we see that improvements are mostly positive, and are much larger for tasks that require a lot of reasoning.

are stacked onto itself to initialize the model's next stage, until the desired depth is reached. This has been shown to significantly speed up BERT pretraining and also has some theoretical justification for the efficiency aspect. While these methods can speed up training, such changes can also induce specific biases into the model. However, the effect of stacking-based approaches on generalization remains a fundamental open question and is largely unexplored.

Modern deep learning models, when trained carefully, have been shown to exhibit interesting inductive biases, and their success is partially attributed to them. Such biases can arise either from model architecture, optimization techniques, or training strategies, and these biases come in various forms including simplicity bias, flatness of learned function, and sparsity. The implicit bias of optimizers, in particular, has been subject to extensive research. For instance, the implicit bias of first-order methods like stochastic gradient descent has been studied extensively in overparametrized settings [Gunasekar et al., 2018, Liu et al., 2023]. Similarly, the inductive biases of architecture components like self-attention and convolution have also been studied [Edelman et al., 2022, Wang and Wu, 2023]. More recently, there has also been interest in constructs like looped models [Lan et al., 2020, Dehghani et al., 2018] that share weights across layers. They have been shown to be powerful enough to emulate programmable computers [Giannou et al., 2023] and have the inductive bias to simulate iterative solutions [Yang et al., 2023], thereby yielding models with algorithmic abilities. However, in this vein, very little is known about the implicit biases of newer training strategies (e.g., greedy layerwise training or gradual stacking) that are gaining popularity.

In this work, we investigate the inductive bias of stacking-based approaches beyond training efficiency. We uncover an intriguing phenomenon — *pretraining with a variant of stacking is not only efficient, but also has a desirable inductive bias towards improving downstream benchmarks*. First, through comprehensive empirical analysis, we discover a novel variant of gradual stacking called MIDAS (MIDdle grAdual Stacking) which copies the middle block of layers of a small network to initialize a larger network (see Figure 1). We demonstrate that MIDAS is more efficient in training compared to standard training and the previous leading stagewise training approach. However, remarkably, it also yields *significantly better performance on many downstream reasoning tasks*. For instance, we see in Figure 1 that MIDAS has significantly better performance on math word problems and reasoning primitives. This performance boost should come as a surprise, since MIDAS uses exactly the same data and fewer training FLOPS compared to standard training. In fact, the pretraining perplexity of MIDAS on a validation set matches that of standard baseline training. This strongly suggests that there is some inductive bias for MIDAS at play.

In this paper, we formalize and provide strong evidence for such an "inductive bias" – MIDAS achieves better downstream evaluations despite performing similarly in terms of pretraining validation perplexity. Thus, the improved quality of MIDAS is not because of better generalization in the pretraining objective, but rather due to its ability to extract more skills and abilities from the pretraining process. This kind of inductive bias phenomenon was first formalized in Saunshi et al. [2022] for contrastive learning and later in Liu et al. [2023] for language modeling on synthetic data. However, this is the first evidence of a strong inductive bias for a training procedure in real language model

training. While our real-world benchmarks already provide strong evidence, in order to better isolate the contributing factors, we construct simple synthetic tasks that are building blocks for reasoning, called *reasoning primitives*. We find that a model pretrained with **MIDAS** has much better performance on the reasoning primitives than a model obtained through standard pretraining, as is evident in Figure 1. In light of the above discussion, we state the main contributions of our paper.

- We propose a novel variant of gradual stacking, called **MIDAS**, that achieves better training efficiency than gradual stacking.

- Our investigation of the inductive bias in gradual stacking approaches, particularly with **MIDAS**, reveals a surprising benefit: *beyond enabling efficient training, it also enhances performance on downstream tasks*. This improvement is especially notable in tasks that rely on context and reasoning abilities.

- We provide strong evidence of the aforementioned phenomenon on several datasets that have previously been used to demonstrate reasoning capabilities.

- We construct simple synthetic tasks that are building blocks for reasoning and demonstrate that **MIDAS** performs significantly better than baseline training on these tasks. These datasets may be of independent interest to the LLM reasoning community.

- Finally, we conjecture the reason behind improved reasoning capabilities of **MIDAS** by presenting connections between gradual stacking and looped models and provide strong empirical evidence to support it.

## 2   Problem Setup

In this section, we first present the problem setup and background material needed for this paper. Before we discuss the problem setting, we set up the following notation for the rest of the paper.

**Notation.** For a deep network $f$, we use $f_i$ and $\#(f)$ to denote the $i^{\text{th}}$ layer and the number of layers of the network, respectively. With slight abuse of notation, we use $f_{i,b}$ (where $i, b \in Z^+$) to denote the layers between $(i-1) \cdot b$ to $i \cdot b$ of a deep network $f$. In other words, $f_{i,b}$ denotes the $i^{\text{th}}$ block of $b$ layers in a deep network $f$. $a_{1:k}$ is used to denote a sequence of $k$ scalars $\{a_1, \ldots, a_k\}$.

Our goal is to learn a function $f : \mathcal{X} \to \mathcal{Y}$ which minimizes the loss $\mathbb{E}_{(x,y) \sim \mathcal{D}} \ell(f(x), y)$, for some loss function $\ell : \mathcal{Y} \times \mathcal{Y} \to \mathbb{R}^+ \cup \{0\}$ and data distribution $\mathcal{D}$ on $\mathcal{X} \times \mathcal{Y}$. We are interested in functions of the form $f = f_L \circ f_{L-1} \circ \cdots \circ f_1$ where $\circ$ and $L$ represent function composition and depth of the network, respectively. We use $\mathcal{F}_L$ to denote the function class consisting of functions of this form. Given samples from the distribution $\mathcal{D}$, we typically use an iterative stochastic optimizer (e.g., SGD) to learn a function that minimizes the loss. We note that the optimization procedure is inconsequential to the arguments in the paper. For standard training, each iteration is of the form:

$$f^t = f^{t-1} + \mathcal{A}(f^{t-1}, \mathcal{B}_t, \eta_t), \qquad \text{(\textbf{Standard Training})}$$

where $\mathcal{B}_t$ is a mini-batch from distribution $\mathcal{D}$ and $\mathcal{A}(f^{t-1}, \mathcal{B}_t, \eta_t)$ represents the iterative optimizer update at $f^{t-1}$ on $\mathcal{B}_t$ and learning rate $\eta_t$. The computation cost and memory requirement for training typically increases linearly with the depth, making even simple algorithms, like SGD, slow for very large models. Throughout this paper, we use $T$ to denote the total number of training iterations.

### 2.1   $k$-stage training

Since we primarily focus on stagewise training approaches, it is useful to formally define a stagewise training procedure. In contrast to standard training, $k$-stage training involves dividing the training process into $k$ stages, and at each stage, using the the model from the previous stage to initialize the model in the current stage. For simplicity, we assume $L$ is divisible by $k$. The following are the key ingredients:

1. **Function class across stages.** At stage $i$, we use function class $\mathcal{F}_{d(i)}$ where $d(i)$ denotes the depth of the network at that stage. When $d(i) \ll L$, training is more efficient.

2. **Training schedules across stages.** As training is divided into $k$ stages, we use $T_1, \cdots, T_k$ steps across stages such that $\sum_{i=1}^{k} T_i = T$.

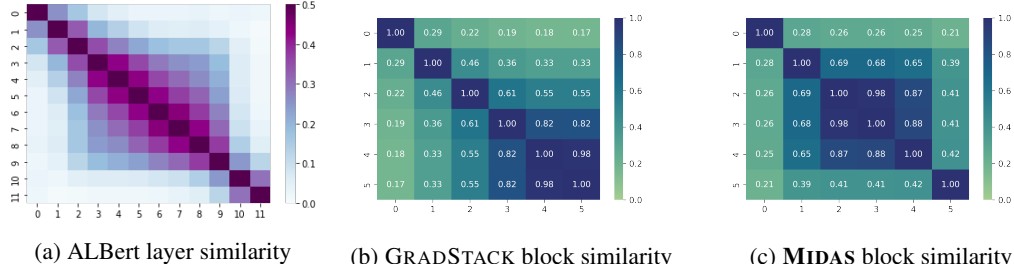

(a) ALBert layer similarity     (b) GRADSTACK block similarity     (c) **MIDAS** block similarity

Figure 2: (a) For an ALBert model trained with weight sharing across all layers, we measure the functional similarity between layers by looking at the top 1% activated neurons in each MLP layer and measure the intersection-over-union (IoU) metric for each pair of layers. Despite all layers having the same parameters, a natural functional similarity structure emerges around the middle. (b) For a UL2 model trained with GRADSTACK, we measure the cosine similarity between every pair of layer blocks for the first feedforward layer weights. (c) The same similarity measured for **MIDAS**. The cosine similarities for stacking based models suggests a strong connection to looped models, and **MIDAS** has a closer similarity structure to ALBert style looped models than GRADSTACK.

3. **Stage initialization.** This is the key component of stagewise training. Given a network $f \in \mathcal{F}_{d(i-1)}$ trained in the $(i-1)^{\text{th}}$ stage, let $\mathcal{M}_i(f)$ denote the network initialization for the next stage where $\mathcal{M}_i : \mathcal{F}_{d(i-1)} \to \mathcal{F}_{d(i)}$ is a growth operator.

Almost all the recent stagewise training procedures are different instantiations of this framework, using different training schedules and stage initializations. We will revisit some prominent instantiations of the framework in the next section.

## 2.2 Progressive & Gradual Stacking

Progressive and gradual stacking are two special instantiations of the aforementioned framework. We provide a brief description of these approaches since they are important for our discussion.

**Progressive Stacking** [Gong et al., 2019]. This is a simple instance of $k$-stage training setup where model in the previous stage is stacked onto itself to initialize the model in the next stage. In particular, **(1)** depth $d(i) = 2^{i-1}d(1)$ grows exponentially, **(2)** schedule $T_i$ is typically $T/k$ or proportional to $d(i)$, and **(3)** the growth function $\mathcal{M}_i(f) = f \circ f$.

**Gradual Stacking** [Reddi et al., 2023]. In contrast to progressive stacking, gradual stacking linearly increases the model depth by $k$ in each stage. It only stacks the last $L/k$ layers of model from the previous stage to initialize the model in the next stage, as follows.

1. The depth $d(i) = \frac{L \cdot i}{k}$ grows linearly with the stage.
2. $T_i$ is typically either $T/k$ or allocated proportional or exponential to depth.
3. $\mathcal{M}_i(f_{d(i-1)} \circ \cdots \circ f_1) = f_{d(i-1)} \cdots \circ f_{d(i-1)-(L/k)+1} \circ f_{d(i-1)} \cdots f_1$. This corresponds to stacking the last $L/k$ layers onto the network to initialize the next stage model.

In the next section, we study a novel variant of gradual stacking that enables faster training and exhibits an interesting inductive bias, which we examine carefully.

## 3 Algorithm: MIDAS

We present the **MIDAS** algorithm in this section. We first discuss the motivation behind this variant of gradual stacking and then formally define the algorithm.

### 3.1 Motivation

The motivation for **MIDAS** touches upon two crucial aspects: (a) the role of different layers in a deep network and (b) a connection to looped models. Before delving into more technical details, it is important to illustrate these points. We present the case for **MIDAS** based on three observations.

**Observation 1: gradual stacking breaks the natural role of layers.** Recall that gradual stacking initializes a larger model by duplicating and stacking the last block of $b$ from the smaller model. Thus in the newly initialized model, the second-last block of $b$ layers will be the same as the last $b$ layers of the smaller model (see Figure 1). Intuitively, this is undesirable since the last few layers have been shown to play a different role compared to other layers for Transformer models [Belrose et al., 2023]. We further validate this in Figure 6. Thus, duplicating the last few layers can break the natural role of layers at the initialization, making it a suboptimal choice. However, it is plausible that the similarity structure across layers is broken after continued training and the initialization is inconsequential. The next observation shows that this is not true, and establishes a connection to looped models – networks with shared parameters between layers.

**Observation 2: gradual stacking leads to models resembling looped models.** To check the effect of the initialization, we measure the cosine similarity between weights of layers for a model pretrained with gradual stacking. In Figure 2b, we observe that indeed the layers continue to have very high cosine similarity at the end of training, thus establishing a connection between stacking and looped models like ALBert [Lan et al., 2020] and Universal Transformers [Dehghani et al., 2018]. Unsurprisingly, the similarity structure for gradual stacking is lopsided towards the end of the model, which raises the question: *Is this similarity structure natural for looped models?*

**Observation 3: looped models exhibit similarity in the middle.** In order to study this, we train a prototypical looped model, ALBert, where all layers share the same parameters. Surprisingly, despite parameters being shared, a natural similarity structure emerges between layers: yet again the first and last layers tend to be functionally dissimilar to other layers, whereas the functional similarity between layers is the highest in the middle (see Figure 2a).

The above observations provides a strong motivation for stacking in the middle rather than at the end, thus inspiring our **MIDAS** algorithm.

### 3.2 MIDAS algorithm

First we define the following mapping operator that is useful for stage initialization in **MIDAS**.

$$\mathcal{M}(f, b) = f_{n,b} \circ \cdots \circ \underbrace{f_{\lceil n/2 \rceil, b} \circ f_{\lceil n/2 \rceil, b}}_{\text{Replication}} \circ \cdots \circ f_{1,b}, \tag{1}$$

where $n = \#(f)/b$ is the number of blocks of $b$ layers in deep network $f$. Note that operator $\mathcal{M}(f, b)$ expands the size of the network by size $b$. Based on this operator, **MIDAS** can again be described as a simple instantiation of the $k$-stage training framework, as seen below. For completeness, the pseudocode for **MIDAS** in listed in Algorithm 1.

---

**Algorithm 1 MIDAS**

**Require:** Schedule $T_{1:k}, \eta_{1:T}$, optimizer update $\mathcal{A}$ (see Section 2), data distribution $\mathcal{D}$.
  **Initialize** $f^{1,0} \in \mathcal{F}_{L/k}$.
  **for** $s = 1 \rightarrow k$ **do**
    **for** $t = 1 \rightarrow T_s$ **do**
      Sample batch $\mathcal{B}_t$ from $\mathcal{D}$.
      $f^{s,t} = f^{s,t-1} + \mathcal{A}(f^{s,t-1}, \mathcal{B}_t, \eta_t)$
    **end for**
    Initializer for next stage:

$$f^{s+1,0} = \mathcal{M}(f^{s,T_s}, L/k)$$

    (see Equation 1)
  **end for**
  **return** $f^{k,T}$

---

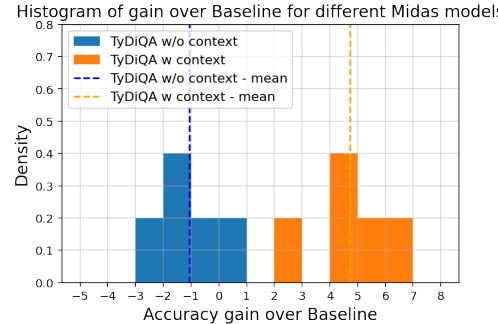

Figure 3: Histogram of accuracy improvements for models trained with **MIDAS** over baseline. The data points are **MIDAS** 1B models listed in Table 1. The figure shows that **MIDAS**-based models have much higher improvement in the contextual version of TyDiQA compared to the non-contextual version.

1. The depth $d(i) = \frac{L \cdot i}{k}$ grows linearly with the stage, similar to gradual stacking.

2. $T_i$ is typically either proportional to $i$ (linear proportional) or $i^2$ (square proportional) or $\exp(i)$ (exponential). We will revisit this during our empirical analysis.

3. We use growth operator $\mathcal{M}$ in equation 1 for initializing the next stage, which corresponds to replicating the middle $L/k$ layers to initialize the next stage model.

## 3.3 Experiments: UL2 Pretraining

In this section, we evaluate **MIDAS** for standard language model pretraining. We train a 24L decoder-only model with 1.5B parameters using the UL2 objective [Tay et al., 2022] on a mixture of C4, Wikipedia, Arxiv and Github. The observations also hold for GPT-style autoregressive language modeling. To enable fair comparison, we cached the pretraining dataset and so all methods are trained for the same number 500B tokens in the same order, using the same batch size (refer to Appendix A.1 for more details on the training setup). We pretrain models with three methods: (a) standard training (*Baseline*), (b) gradual stacking (GRADSTACK) and (c) our proposed method **MIDAS**. The goal is to compare them with respect to validation loss and downstream performance on several diverse benchmarks. Motivated by the proportional schedules from prior work, we try the following generalized proportional schedules for gradual stacking and **MIDAS**.

**Definition 3.1** (PROP-$\alpha$ schedule). *For a total training budget of $T$ steps, the schedule PROP-$\alpha$ spends time $T_i$ in each stage such that $T_i \propto i^\alpha$ for all stages $i \in [k]$. Thus $T_i = \frac{i^\alpha}{\sum_{j=1}^{k} j^\alpha} T$*

PROP-1 schedule has been found to work very well for BERT pretraining [Reddi et al., 2023]. Since UL2 pretraining is a harder task, we also explore less aggressive schedules like PROP-2 and PROP-3 that spend more time on larger models.

**Efficiency and perplexity findings.** We summarize the main results in Table 1, for various stacking methods and schedules. Firstly, we note that for all schedules, **MIDAS** has significantly better validation log perplexity than GRADSTACK at the same speedup level. This suggests that stacking in the middle is a lot more effective for optimization than stacking at the end of the model. With the PROP-2 schedule, **MIDAS** is 24% faster and nearly matches the baseline's log perplexity. Additionally, we observe that the findings are robust to the choice of block size for stacking.

**Downstream benchmark evaluations.** While perplexity can serve as a decent proxy for model quality, there is growing evidence that it is not the best measure [Liang et al., 2023]. Downstream benchmark evaluations serve as a more holistic measure for quality and are out-of-distribution evaluations of skills. To this effect, we evaluate **MIDAS** on many standard benchmarks and these are grouped into task categories in Table 1 (refer to Appendix A.2 for more detailed evaluations on individual tasks). The accuracy for task category is an average over representative tasks from that group. For instance, for closed book QA task, we consider an average accuracy on TriviaQA, TydiQA (no context), NaturalQuestions and WebQuestions.

Surprisingly, we find that downstream improvements for **MIDAS** are significantly larger than the improvements in perplexity. In particular, **MIDAS** with PROP-2 schedule has very similar perplexity to baseline at 24% speedup, but the average downstream performance for **MIDAS** (26.8%) is much better than baseline (24.0%). In fact, even **MIDAS** with PROP-1 schedule which has worse log perplexity is much better on downstream evaluations. Similar trends of better downstream evals holds for the 2B parameter model. The improvements are particularly large for open book QA and math word problems, both of which are tasks that require reasoning abilities whereas memorization tasks like closed book QA do not improve. We conjecture that these downstream improvements are due to an *inductive bias* induced by stacking and we dive deeper into this in the next section.

## 4 Inductive bias of stacking

Results in Table 1 demonstrate that **MIDAS** not only yields training speedups, but also improves downstream evaluations when trained on the same number of tokens as standard training. This suggests that stacking can extract more *skills* out of the same data. Here, we take a closer look at these improvements in downstream evaluations through the lens of an *inductive bias* of stacking.

Table 1: Downstream evaluations for UL2 pretrained models with 1B, 2B and 8B parameters. Comparisons include standard training (Baseline), gradual stacking (GRADSTACK) from [Reddi et al., 2023] and our proposed method MIDAS. The downstream evaluations are averaged over tasks within 3 task groups. See Appendix A for precise tasks included in each task group. For each cateory and model size, we highlight the top model is **bolded** and the second best model is underlined. Firstly, MIDAS is much better than GRADSTACK, thus justifying stacking in the middle. Secondly, MIDAS can match the log perplexity of baseline training while being roughly 24% faster. Furthermore, even the schedule with 40% speedup has much better downstream evaluations compared to baseline, even though it has worse log perplexity. The improvements are particularly large for task groups that require reasoning (open book QA, math word problems).

| $d(i)/i$ (block size) | Schedule | Speedup | Loss ($\downarrow$) (validation) | Closed Book QA ($\uparrow$) (4 tasks) | Open Book QA ($\uparrow$) (5 tasks) | Math Word Problems ($\uparrow$) (6 tasks) | All Tasks Average ($\uparrow$) (15 tasks) |
|---|---|---|---|---|---|---|---|
| **1B Parameters** | | | | | | | |
| Baseline | 24 | | 1x | **1.996** | **13.2** | 33.3 | 23.5 | 24.0 |
| GRADSTACK | 4 | PROP-1 | 1.39x | 2.045 | 10.3 | 31.4 | 23.5 | 22.6 |
| MIDAS | 4 | PROP-1 | 1.39x | 2.028 | 11.6 | 34.5 | 30.3 | 26.7 |
| MIDAS | 3 | PROP-1 | 1.41x | 2.032 | 10.6 | 36.1 | 27.0 | 25.6 |
| GRADSTACK | 4 | PROP-2 | 1.24x | 2.024 | 11.0 | 31.6 | 17.3 | 20.4 |
| MIDAS | 4 | PROP-2 | 1.24x | 2.009 | 11.7 | 36.3 | 29.0 | 26.8 |
| MIDAS | 3 | PROP-2 | 1.26x | 2.012 | 11.9 | **37.3** | 29.8 | 27.5 |
| MIDAS | 4 | PROP-3 | 1.16x | 1.999 | 12.5 | 34.8 | **33.3** | **28.3** |
| **2B Parameters** | | | | | | | |
| Baseline | 48 | | 1x | **1.926** | 15.2 | 39.1 | 27.1 | 28.0 |
| MIDAS | 8 | PROP-1 | 1.39x | 1.947 | 14.0 | 38.9 | 32.0 | 29.5 |
| GRADSTACK | 8 | PROP-2 | 1.24x | 1.945 | 14.2 | 37.0 | 24.5 | 25.9 |
| MIDAS | 8 | PROP-2 | 1.24x | 1.929 | **15.7** | **40.2** | **38.2** | **32.9** |
| **8B Parameters** | | | | | | | |
| Baseline | 72 | | 1x | **1.841** | 21.1 | 39.6 | 34.9 | 32.8 |
| MIDAS | 9 | PROP-2 | 1.26x | 1.844 | **21.8** | **40.0** | **43.1** | **36.4** |

## 4.1 Downstream performance vs log perplexity

A reasonable expectation from pretraining is that improvements in the pretraining objective would correlate with improvements in model quality and downstream performance. This notion of transfer has even been theoretically formalized for language modeling in Saunshi et al. [2020], Arora and Goyal [2023]. Thus, based on this, a natural explanation for the downstream improvements of stacking would be that it generalizes better on the pretraining objective. However, as we see in Table 1, downstream performance of MIDAS is better despite having similar or worse *validation* perplexity – hence this is not simply the case of better generalization to unseen pretraining data. It is natural to ask: *If not perplexity, what explains this downstream phenomenon?*

Since pretraining objective is just a proxy objective for model quality, it is plausible that different training strategies and model architectures can extract different levels of skills from it. This is because there are multiple ways of doing well on the pretraining tasks, and some training strategies can be biased to pick one solution over another one. This behavior has been formalized as the inductive bias in pretraining by recent work [Saunshi et al., 2022, Liu et al., 2023] – at the same level of validation pretraining loss, different optimization algorithms could have vastly different downstream performance. We hypothesize that a similar phenomenon is at play when it comes to stacking.

**Isoplots.** Inspired by this phenomenon of different downstream performance at the same perplexity, we visualize the inductive bias of a method by plotting downstream accuracy vs log perplexity isoplots as training proceeds. We use the UL2 1B models that are pretrained with standard (baseline) training and with MIDAS using the PROP-2 schedule (refer to Section 3.3 for more details). In Figure 4, we visualize the downstream vs log perplexity plots for different task groups – closed-book QA, open-book QA and math word problems. We observe a very interesting trend – MIDAS and baseline training can have different isoplot behaviors and the divergence is different for different tasks.

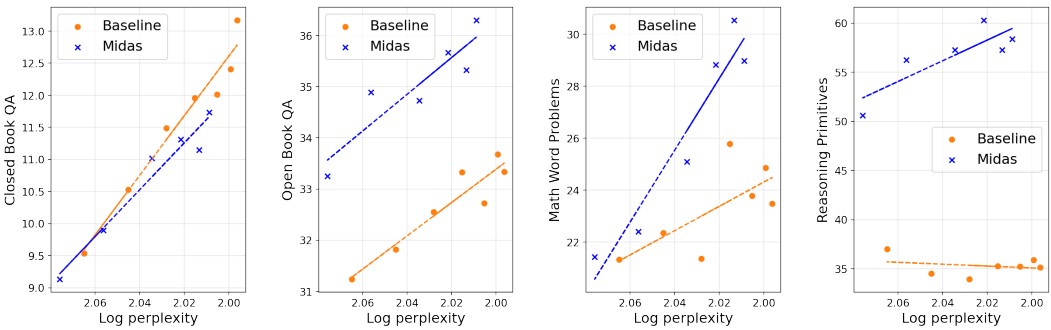

Figure 4: Downstream evaluation vs validation log perplexity isoplots as training proceeds for baseline and **MIDAS** 1B models trained on the same data (stacking is 24% faster here). On the y-axis we track the performance on various task groups – closed book QA, open book QA, math word problems and our reasoning primitives from Section 5. On the x-axis the log perplexity is presented in the reverse order, thus downstream performance for both methods improves as log perplexity gets lower. For closed book QA (memorization) tasks **MIDAS** has very similar trends to baseline. For open book QA tasks and math word problems, **MIDAS** has much better downstream performance at an equivalent log perplexity. This showcases the inductive bias of **MIDAS** towards better overall quality and better reasoning abilities.

## 4.2 Reasoning vs memorization for QA

For a clearer display of the inductive bias, we measure the improvements due to **MIDAS** on closed book vs open book QA tasks. It is reasonable to assume that closed book QA tasks require strong memorization abilities whereas open book QA tasks require some reasoning abilities to infer answers from the context that is provided. On average, we see much larger improvements on open book QA tasks compared to closed book QA tasks, as already evident in Figure 1 and Table 1.

**MIDAS is significantly better on Open book QA.** To make a direct comparison, we consider TydiQA-GoldP and TydiQA-NoContext tasks – the datasets are identical and the only difference is whether or not additional context is provided (the answer for the contextual version is guaranteed to be inferred from the given context). In Figure 3, we see that the improvements by various **MIDAS** based models on the contextual version of TydiQA are much higher than those on the non-contextual version. This provides direct evidence of the bias of **MIDAS** towards improving tasks that require reasoning. Furthermore, we find that the memorization performance of stacking improves as the schedule spends more time on the larger model.

## 4.3 Reasoning in math tasks

To test reasoning abilities, we evaluate the language models on various math word problem datasets like SVAMP [Patel et al., 2021], ASDiv [Miao et al., 2020], AQuA dataset for algebraic word problems, the MAWPS benchmark [Koncel-Kedziorski et al., 2016]. We report 5-shot evaluation for the pretrained model on these tasks. Following Wei et al. [2022], we use an external calculator to do the arithmetic and evaluate the models on their ability to compute the correct expression for the answer. This is because small models have bad arithmetic accuracy. The choice of using calculator or not does not significantly affect the trends of the results. For stacking, we use **MIDAS** PROP-2 model because it achieves nearly the same perplexity as the baseline model (while being 24% faster), thus, leading to a fair comparison based on the previous notion of inductive bias.

**MIDAS is significantly better on Math/Reasoning tasks.** Detailed results can be found in Table 5. For most math tasks, we observe that the **MIDAS**-based pretrained model is significantly better than the baseline model, especially for the MAWPs benchmark. This provides further evidence of better math and reasoning capabilities of **MIDAS**.

**GSM8K fine-tuning.** We also evaluate the 2B and 8B models on harder math problems from the GSM8k dataset [Cobbe et al., 2021] through few-shot prompting and fine-tuning. Full results are presented in Table 2. For **MIDAS** we use the PROP-2 model that has very similar perplexity as the

Table 2: Evaluation on math tasks, including math word problems from Table 1 and a harder task GSM8k. For GSM8k we report accuracy with 8-shot prompts and with finetuning. We also report accuracy on all tasks after using an external calculator to fix arithmetic errors; this corresponds to w/ calc. Overall the use of calculator improves the accuracy for all models on all tasks. The benefit of **MIDAS** over baseline is even higher with calculator.

| Model | Pretraining Loss (↓) | Math WPs (5-shot) | | GSM8k (8-shot) | | GSM8k (Finetune) | |
|---|---|---|---|---|---|---|---|
| | | W/o calc. | W calc. | W/o calc. | W calc. | W/o calc. | W calc. |
| **2B Parameters** | | | | | | | |
| Baseline | 1.926 | 15.4 | 27.1 | 3.0 | 3.6 | 5.3 | 8.5 |
| **MIDAS** | 1.929 | 22.5 | 38.3 | 3.0 | 4.1 | 10.4 | 14.5 |
| **8B Parameters** | | | | | | | |
| Baseline | 1.841 | 27.3 | 34.9 | 4.5 | 6.6 | 12.3 | 15.8 |
| **MIDAS** | 1.844 | 32.9 | 43.1 | 5.5 | 7.4 | 15.2 | 18.7 |

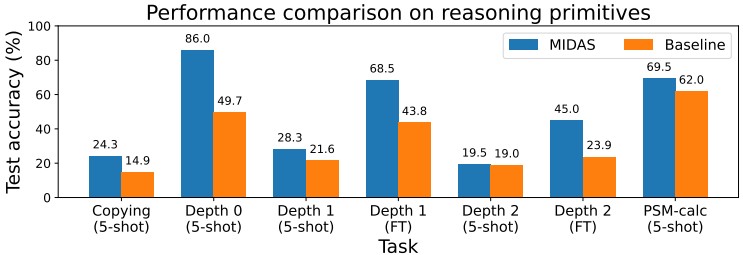

Figure 5: Accuracy improvements for model trained with **MIDAS** over baseline for representative *reasoning primitives*, despite having the same perplexity. We see clear improvements for **MIDAS** on almost all the primitives, both with 5-shot evaluation and after fine-tuning (FT) for the depth 1 and 2 primitive.

baseline model. We find that **MIDAS** has much higher accuracy after fine-tuning, thus suggesting that the benefits of the inductive bias continue after fine-tuning and are not just restricted to few-shot evaluations. In particular, on the test set, the accuracy metric increased from 5.3% (for the baseline model) to 10.4% (for **MIDAS**) for the 2B model (these numbers were produced by computing the average score over three runs with different random seeds). Similarly the GSM8k accuracy of the 8B model improves from 12.3% to 15.2%. This suggests that **MIDAS** not only improves the performance on harder math tasks, but also that the gains remain or improve after fine-tuning.

**Effect of calculator.** For LLMs with less than 20B parameters, Wei et al. [2022] found that models often solve math problems correctly but make arithmetic errors, leading to low accuracy. Wei et al. [2022] remedied this by computing all arithmetic expressions using a Python program as an external calculator. In Table 2 we find that this improves the accuracy for our models too. Interestingly, the gap between **MIDAS** and baseline gets even larger with calculator use in almost all comparisons. We believe this is because arithmetic abilities are closer to memorization for smaller models [Razeghi et al., 2022] and calculator use makes the problem closer to reasoning, since now the model only has to infer the right expression. We believe this interplay between reasoning and memorization for math problems deserves further investigation.

## 4.4 Connection to looped models

Given the nature of the growth operator in each stage, we hypothesize that stacking based models are close to looped models. The layer duplication that happens at every stage ensures that blocks of layers start from a common initialization. We measure the similarity between different blocks of layers by measuring cosine similarities between the parameter vectors (see Figure 2). Since looped models have been conjectured to solve algorithmic problems [Giannou et al., 2023] by finding iterative solutions [Yang et al., 2023], we conjecture that the better reasoning abilities of **MIDAS** are due to this connection to looped models We believe exploring this further is a very fruitful direction.

# 5 Deep dive into reasoning improvements

To further investigate the nature of this inductive bias, we construct various simple synthetic tasks to help tease apart the model's capabilities. We conjecture that these simple tasks capture core basic capabilities needed for contextual reasoning, and we therefore call these tasks "contextual reasoning primitives". They are: induction copying, variable assignment, and pre-school math (PSM), discussed further below. Overall, across various few-shot evaluations and fine-tuning, we see significant performance gaps between **MIDAS** and baseline training, suggesting that we have successfully isolated some of the basic capabilities at which **MIDAS** excels relative to baseline training. We refer the reader to Appendix B for more results and the exact input format.

**Primitive 1: Induction copying.** The "induction copying" primitive presents a sequence of words, followed by a subsequence selected randomly from within this original sequence, and asks the model to output the *next* word in the sequence. A simplified example is: "pum nyj gdq ocu `rzk jbw mlz eny kyx uni rzk jbw mlz eny kyx`", and the expected output is "uni". This primitive is inspired by the "induction head" mechanism introduced in Olsson et al. [2022], which is posited to be the basic mechanism for in-context learning more generally. In Figure 5, task "Copying", we present results for 3-letter words of random letters, separated by spaces, with a sequence length of 10 and a subsequence length of 5.

**Primitive 2: Variable assignment.** The "variable assignment" primitive tests the model's ability to associate a value with a variable name and apply this ability *compositionally*, which we test by varying the "depth" of the task. We conjecture that this ability is a core function in contextual reasoning, particularly in math. An example of the depth-0 variant is "u=1; t=0; v=13; y=4; f=22; y=", and the expected output is 4. An example of the depth-2 variant is "y=7; f=0; z=3; b=9; x=8; q=y; l=f; m=z; h=x; a=b; n=h; j=m; t=a; i=l; g=q; n=", and the expected output is 8. Refer to Appendix B for more details.

**Primitive 3: Pre-school math (PSM).** This tests the model's ability to solve a very simple "pre-school math" problem by correctly associating multiple values and variables *simultaneously* and applying this association to a particular task. An example is "z=6; b=5; i=-z+b; i=", and the expected answer (with chain-of-thought) is "-6+5=-1".

**5-shot evaluation results.** Figure 5 presents the results for representative tasks, with more results in Appendix B. Overall, we see that **MIDAS** outperforms baseline training across all tasks. In particular, we see that **MIDAS** is significantly stronger than the baseline at Depth 0, Copying, PSM-calc, and Depth 1, in decreasing order of magnitude of the performance gap. Depth-2 is much harder and is at random guessing (20%) for both models.

**Fine-tuning results.** Due to the difficulty of the variable assignment task at Depths 1 and 2, we investigate fine-tuning on these tasks as well. We fine-tune on a mixture of 32 depth-1 examples and 32 depth-2 examples (i.e., only 64 examples total), using full-batch gradient descent. Figure 5 reports the validation accuracy on Depth 1 and Depth 2 after fine-tuning on this mixture (tasks "Depth 1 (FT)" and "Depth 2 (FT)"). Overall, we see that fine-tuning with just 64 examples significantly improves performance, resulting in **MIDAS** outperforming the baseline by a gap of over 20% validation accuracy at both depths. See Appendix B for further fine-tuning and evaluation details.

# 6 Conclusions and future work

In this work we propose a novel stacking method that outperforms previous stacking methods and speeds up language model pretraining by 25-40%. In the process, we uncover a very intriguing inductive bias of stacking – its ability to improve downstream reasoning tasks. Through extensive empirical analysis, the paper makes a strong case for the presence and significance of this inductive bias. We believe this deserves further attention and exploration since understanding this inductive bias could unlock new approaches to improving model quality, reasoning in particular. The reasoning primitives start to provide more insights by isolating the reasoning improvements and we hope that the dataset is useful for future research on improving reasoning. Finally, understanding the dichotomy between memorization and reasoning, and how this affects the performance on various tasks, is an interesting direction to pursue.

**Acknowledgments.** We thank Srinadh Bhojanapalli and Vaishnavh Nagarajan for discussions on the role of layers and memory vs contextual tasks, respectively, in the early stages of the project. We also thank Satyen Kale for valuable feedback throughout the project.

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

# A Experimental Details

## A.1 Pretraining details

**Model architecture.** We use a decoder-only model and train it using the UL2 objective [Tay et al., 2022] with 60% causal LM, 20% prefix LM and 20% span corruption. The 1B model uses 24 layers, model dimension of 2048, hidden dimension of 5120 and 32 attention heads. The 2B model is very similar to the 1B model, except it uses 48 layers instead of 24. The 8B model uses 72 layers, model dimension of 2048, hidden dimension of 16384 and 16 attention heads.

**Dataset.** We use a mixture of C4 (57%) [Raffel et al., 2020], Wikipedia (17%), Github (17%), Arxiv (9%); the proportions are motivated by the dataset used for Llama pretraining [Touvron et al., 2023]. All models are trained for 512B tokens that are precached so that all model see exactly the same data in the same order. This corresponds to 0.86 epochs of C4, 9 epochs of Wikipedia, 0.58 epochs of Arxiv and 0.44 epochs of Github.

**Training details.** For the 1B and 2B models, we use a cosine learning schedule with a peak learning rate of $0.01$ that decays to $0.001$ in the end, and use a batch size of 512. For the 8B model we use a peak learning rate of $0.001$ and decay it to $0.0001$, and use a batch size of 1024. Peak learning rate was tuned to be optimal for baseline training. All experiments use the AdaFactor optimizer [Shazeer and Stern, 2018] and sequence length of 1280.

## A.2 Additional downstream evaluations

In this section we share further experimental details related to the results summarized in the Table 1.

| | $d(i)/i$ | Schedule | Speedup | Trivia QA | TyDi QA (w/o Context) | Natural Questions | Web Questions |
|---|---|---|---|---|---|---|---|
| **1B Parameters** | | | | | | | |
| Baseline | 24 | | 1x | 28.1 | 12.0 | 4.5 | 8.1 |
| GRADSTACK | 4 | PROP-1 | 1.39x | 22.4 | 10.1 | 3.0 | 5.8 |
| **MIDAS** | 4 | PROP-1 | 1.39x | 25.0 | 11.7 | 3.7 | 5.9 |
| **MIDAS** | 3 | PROP-1 | 1.41x | 22.9 | 9.6 | 3.5 | 6.5 |
| GRADSTACK | 4 | PROP-2 | 1.24x | 22.9 | 11.4 | 4.0 | 5.9 |
| **MIDAS** | 4 | PROP-2 | 1.24x | 26.4 | 10.4 | 3.7 | 6.4 |
| **MIDAS** | 3 | PROP-2 | 1.26x | 25.5 | 10.9 | 3.8 | 7.4 |
| **MIDAS** | 4 | PROP-3 | 1.16x | 26.9 | 12.0 | 4.5 | 6.8 |
| **2B Parameters** | | | | | | | |
| Baseline | 48 | | 1x | 33.6 | 12.8 | 5.9 | 8.7 |
| **MIDAS** | 8 | PROP-1 | 1.39x | 31.1 | 11.7 | 5.6 | 7.8 |
| GRADSTACK | 8 | PROP-2 | 1.24x | 32.0 | 12.5 | 5.8 | 6.7 |
| **MIDAS** | 8 | PROP-2 | 1.24x | 34.6 | 13.0 | 6.3 | 8.9 |
| **8B Parameters** | | | | | | | |
| Baseline | 72 | | 1x | 47.0 | 15.2 | 9.6 | 12.9 |
| **MIDAS** | 9 | PROP-2 | 1.26x | 47.9 | 17.0 | 9.2 | 13.1 |

Table 3: Closed Book QA

| | $d(i)/i$ | Schedule | Speedup | TyDi QA (w/ Context) | SquadV2 | DROP | QuAC | CoQA |
|---|---|---|---|---|---|---|---|---|
| **1B Parameters** | | | | | | | | |
| Baseline | 24 | | 1x | 31.4 | 41.1 | 22.9 | 18.8 | 52.6 |
| GRADSTACK | 4 | PROP-1 | 1.39x | 34.3 | 36.9 | 21.5 | 17.5 | 46.8 |
| **MIDAS** | 4 | PROP-1 | 1.39x | 36.1 | 39.1 | 24.3 | 18.7 | 54.4 |
| **MIDAS** | 3 | PROP-1 | 1.41x | 37.0 | 44.9 | 25.0 | 18.4 | 55.1 |
| GRADSTACK | 4 | PROP-2 | 1.24x | 30.0 | 41.0 | 22.1 | 17.2 | 47.8 |
| **MIDAS** | 4 | PROP-2 | 1.24x | 35.5 | 46.6 | 24.4 | 19.7 | 55.4 |
| **MIDAS** | 3 | PROP-2 | 1.26x | 38.2 | 46.3 | 24.8 | 19.9 | 57.3 |
| **MIDAS** | 4 | PROP-3 | 1.16x | 33.6 | 40.2 | 24.7 | 19.5 | 55.9 |
| **2B Parameters** | | | | | | | | |
| Baseline | 48 | | 1x | 42.5 | 49.6 | 25.1 | 20.6 | 57.8 |
| **MIDAS** | 8 | PROP-1 | 1.39x | 37.7 | 48.9 | 26.1 | 20.1 | 61.8 |
| GRADSTACK | 8 | PROP-2 | 1.24x | 38.0 | 47.9 | 23.6 | 19.0 | 56.7 |
| **MIDAS** | 8 | PROP-2 | 1.24x | 41.8 | 48.0 | 27.9 | 20.7 | 62.6 |
| **8B Parameters** | | | | | | | | |
| Baseline | 72 | | 1x | 39.1 | 51.8 | 25.9 | 19.6 | 61.6 |
| **MIDAS** | 9 | PROP-2 | 1.26x | 38.9 | 48.9 | 27.0 | 20.5 | 64.8 |

Table 4: Open Book QA

| | $d(i)/i$ | Schedule | Speedup | ASDiv | MAWPS Add/Sub | MAWPS Multi-Arith | MAWPS Single-Eq | MAWPS Single-Op | SVAMP |
|---|---|---|---|---|---|---|---|---|---|
| **1B Parameters** | | | | | | | | | |
| Baseline | 24 | | 1x | 21.7 | 39.0 | 1.7 | 30.5 | 34.2 | 13.9 |
| GRADSTACK | 4 | PROP-1 | 1.39x | 19.1 | 38.8 | 2.0 | 31.1 | 35.2 | 15.1 |
| **MIDAS** | 4 | PROP-1 | 1.39x | 27.7 | 45.1 | 2.8 | 40.2 | 49.1 | 16.9 |
| **MIDAS** | 3 | PROP-1 | 1.41x | 25.8 | 45.1 | 2.5 | 33.1 | 40.7 | 14.8 |
| GRADSTACK | 4 | PROP-2 | 1.24x | 15.2 | 29.1 | 1.0 | 24.6 | 26.3 | 7.6 |
| **MIDAS** | 4 | PROP-2 | 1.24x | 26.3 | 51.9 | 3.3 | 39.4 | 40.0 | 13.0 |
| **MIDAS** | 3 | PROP-2 | 1.26x | 28.6 | 39.0 | 3.0 | 41.1 | 50.4 | 16.8 |
| **MIDAS** | 4 | PROP-3 | 1.16x | 28.9 | 55.7 | 1.5 | 41.1 | 50.9 | 21.8 |
| **2B Parameters** | | | | | | | | | |
| Baseline | 48 | | 1x | 27.9 | 41.5 | 3.2 | 37.4 | 36.5 | 16.4 |
| **MIDAS** | 8 | PROP-1 | 1.39x | 29.0 | 56.2 | 1.0 | 41.9 | 45.9 | 18.1 |
| GRADSTACK | 8 | PROP-2 | 1.24x | 22.7 | 43.0 | 3.2 | 30.5 | 33.1 | 14.3 |
| **MIDAS** | 8 | PROP-2 | 1.24x | 34.7 | 58.2 | 7.3 | 50.0 | 57.5 | 21.8 |
| **8B Parameters** | | | | | | | | | |
| Baseline | 72 | | 1x | 35.0 | 44.6 | 3.7 | 46.0 | 57.1 | 22.9 |
| **MIDAS** | 9 | PROP-2 | 1.26x | 39.3 | 60.8 | 5.2 | 54.9 | 66.0 | 32.2 |

Table 5: Math World Problems

# B  Details for contextual reasoning primitives

In this section, we provide further details corresponding to Section 5.

All evaluations in Section 5 were performed on the 1B-parameter models. For **MIDAS**, we use the variant with block size 4 and the PROP-2 schedule.

## B.1  Exact input format

Expanding on Section 5, here we provide the format of the inputs and target outputs. The only caveat is that, for simplicity of presentation, we present the inputs in 0-shot form here vs. their 5-shot form. In 5-shot form, which is how we conduct the 5-shot evaluations, each example is separated by two consecutive newline characters.

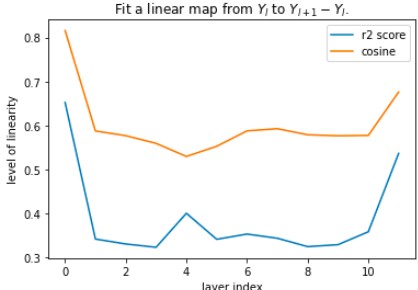 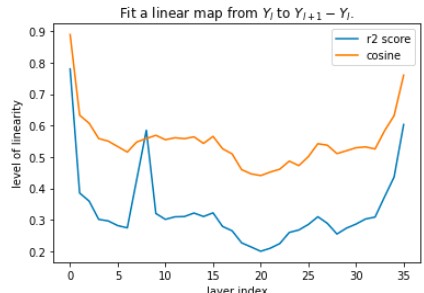

Figure 6: Measure of linearity for different layers in pretrained BERT-Base and BERT-Large models. For each layer $i$, we fit a linear map $A_i$ between inputs $Y_i$ and the output of the Transformer block (without the residual connection), $Y_{i+1} - Y_i$. We then measure the r2 score and cosine similarity for the learned linear fit. The first and last few layers demonstrate a much higher level of linearity compared to the rest of the layers.

For each dataset below, the inputs are separated from the targets by the "|" character (this is not a token in the input), and the targets are colored in red.

Figure 5 uses the following *evaluation* datasets, in the following order:

1. Copying (random-letter words)
2. Variable assignment depth 0 (code)
3. Variable assignment depth 1 (code)
4. Variable assignment depth 1 (code)
5. Variable assignment depth 2 (code)
6. Variable assignment depth 2 (code)
7. Pre-school math (PSM)

---

**Copying (random-letter words):**

```
Fill in blank:

pum nyj gdq ocu rzk jbw mlz eny kyx uni rzk jbw mlz eny kyx ___.   ->|uni
```

---

**Copying (real words):**

```
Fill in blank:

eat fit ban sea vet zit pea cat van tea sea vet zit pea cat ___.   ->|van
```

---

**Variable assignment depth 0 (basic):**

```
Fill in blank:

o=14
s=4
u=8
m=10
q=12
m=___.   ->|10
```

## Variable assignment depth 1 (basic):

```
Fill in blank:

g=21
b=24
v=3
s=23
h=20
k=b
a=s
n=v
f=g
d=h
a=___.   ->|23
```

## Variable assignment depth 2 (basic):

```
Fill in blank:

w=24
l=12
d=16
e=5
j=9
g=j
y=e
r=l
k=d
h=w
v=g
i=r
c=h
t=k
p=y
c=___.   ->|24
```

## Variable assignment depth 0 (math):

```
The following is a set of simple mathematical equations.
n=22
r=16
w=13
v=6
k=10
What is the numerical value of n?
Answer:|22
```

---

**Variable assignment depth 1 (math):**

```
The following is a set of simple mathematical equations.
h=20
w=9
c=22
j=11
v=5
g=c
k=w
a=j
s=h
o=v
What is the numerical value of s?
Answer:|20
```

---

**Variable assignment depth 2 (math):**

```
The following is a set of simple mathematical equations.
g=9
v=24
k=15
p=6
c=10
t=p
s=g
a=c
y=v
n=k
l=s
w=n
j=t
m=y
i=a
What is the numerical value of j?
Answer:|6
```

---

**Variable assignment depth 0 (code):**

```
The following is a very short Python program.  Use the program to resolve
the value of the variable in the question.

Program:
q=12
k=17
l=1
y=3
a=6

Question:
What is the value of k?

Answer:
|17
```

---

**Variable assignment depth 1 (code):**

```
The following is a very short Python program.  Use the program to resolve
the value of the variable in the question.

Program:
k=11
f=21
e=10
l=7
c=13
y=f
o=c
r=e
u=k
n=l

Question:
What is the value of o?

Answer:
|13
```

---

**Variable assignment depth 2 (code):**

```
The following is a very short Python program.  Use the program to resolve
the value of the variable in the question.

Program:
t=13
j=14
v=4
s=17
y=21
q=j
l=s
```

```
e=y
h=t
x=v
b=x
f=e
n=q
a=h
i=l

Question:
What is the value of i?

Answer:
|17
```

---

**Pre-school math (PSM):**

```
Fill in blank:

k=1
j=8
l=-k+j
l=___.   ->|-1+8=7
```

---

**Arithmetic:**

```
-3+2=-1

-6+1=-5

+9-7=2

-6-4=-10

-6-1=-7

+1+9=|10
```

---

## B.2   Fine-tuning details

For fine-tuning, we use the "code" variant of the variable assignment task, depths 1 and 2, in 0-shot form (i.e., no in-context examples). Due to the randomness of the data generation process and the rather small size of each dataset (64 examples), we randomly generate 3 different 64-example fine-tuning datasets (consisting of 32 depth-1 examples and 32 depth-2 examples), fine tune on each, and report our results as an average across the 3 runs. Table 7 reports the standard deviations as well.

Regarding hyperparameters, we continue to use AdaFactor [Shazeer and Stern, 2018] with the same hyperparameters as in the pretraining phase, with the exception of learning rate and batch size. We use

a constant learning rate of 0.001, which was chosen to match the final learning rate of the pretraining phase. We use full-batch training with our 64-example datasets. We then evaluate performance separately on depth 1 and depth 2.

For every step $i \in \{200, \ldots, 300\}$, chosen to be significantly *after* training has converged to 100% accuracy (we do not observe overfitting in this range as training continues), we evaluate performance on a 1000-example holdout set. For smoothing purposes, we average over steps 200 through 300 and report the final averaged performance.

## B.3 Full 5-shot and fine-tuning results

**5-shot.** Table 6 includes 5-shot evaluation results for all contextual reasoning primitives. Rows 1, 9, 10, 11, and 14 are the rows which appear in Figure 5.

When performance is better than random guessing, **MIDAS** consistently outperforms the baseline in rows 1-11.

For pre-school math (rows 12-14), the value we report in Figure 5 is "with calculator". This is because the pre-school math task actually combines two capabilities: reasoning and arithmetic. Arithmetic can be thought of as a memorization task. We evaluate arithmetic for **MIDAS** and baseline training, and we see that arithmetic is quite poor for both models (7.8% and 9.6%, respectively, in Table 6). However, by evaluating PSM with chain-of-thought and only assessing the accuracy of the reasoning chain itself, i.e., "-6+5" vs. "-1", we can successfully disentangle reasoning and memorization in our evaluation. This is equivalent to having access to a calculator, so we call it "PSM with calculator" or "PSM-calc" in Figure 5.

| Task | MIDAS (%) | Baseline (%) | Random guessing(%) |
|---|---|---|---|
| Copying (random-letter words) | 24.3 | 14.9 | 10 |
| Copying (real words) | 17.8 | 10.3 | 10 |
| Variable assignment depth 0 (basic) | 35.6 | 32.1 | 20 |
| Variable assignment depth 1 (basic) | 20.6 | 21.9 | 20 |
| Variable assignment depth 2 (basic) | 18.9 | 17.7 | 20 |
| Variable assignment depth 0 (math) | 92.8 | 50.1 | 20 |
| Variable assignment depth 1 (math) | 26.5 | 19.2 | 20 |
| Variable assignment depth 2 (math) | 20.4 | 18.8 | 20 |
| Variable assignment depth 0 (code) | 86.0 | 49.7 | 20 |
| Variable assignment depth 1 (code) | 28.3 | 21.6 | 20 |
| Variable assignment depth 2 (code) | 19.5 | 19 | 20 |
| Pre-school math (PSM), no calculator | 7.8 | 9.6 | n/a |
| Arithmetic-only accuracy | 9.7 | 10.3 | n/a |
| Pre-school math (PSM), with calculator | 69.5 | 62 | n/a |

Table 6: 5-shot results for all variants of the contextual reasoning primitives. This is an expanded set compared to Figure 5.

**Fine tuning.** Table 7 presents the fine-tuning results from Figure 5 along with corresponding standard deviations (across the 3 trials).

| Task | MIDAS (%) | Baseline (%) | Random guessing(%) |
|---|---|---|---|
| Variable assignment depth 1 (code) | $68.54 \pm 7.69$ | $43.75 \pm 5.54$ | 20 |
| Variable assignment depth 2 (code) | $44.97 \pm 7.26$ | $23.88 \pm 1.56$ | 20 |

Table 7: Fine-tuning results corresponding to Figure 5's 2 fine-tuning tasks. Additionally, this table reports the standard deviation across the 3 runs with $\pm$ std dev.

