# OpenReview forum: "On the Inductive Bias of Stacking Towards Improving Reasoning"
_NeurIPS.cc/2024/Conference — NeurIPS 2024 poster_

### Official Review · Reviewer_s9u7 · 2024-07-12

**Soundness:** 3
**Presentation:** 4
**Contribution:** 3
**Rating:** 7
**Confidence:** 4

**Summary:**

This paper examines the inductive bias of gradually stacking layers to increase the depth of a smaller model. The proposed stacking variant, MIDAS, enhances training efficiency and discovers a compelling inductive bias that boosts downstream performance, particularly in reasoning tasks.

**Strengths:**

1) This work is well-motivated in its aim to design a strategy for reducing the computational cost of training large models. Like some previous studies, it proposes a straightforward variant of gradual stacking that results in improved training speed and performance on reasoning tasks.
2) They established a link between the surprisingly discovered inductive bias that enhances reasoning performance and the looped transformer, which is specifically designed for such tasks. This work paves the way for further research into understanding the intriguing inductive bias of stacking layers to grow model size, not only for computational gain but also for performance improvement.
3) The paper is well written.

**Weaknesses:**

The benchmarked models of size 1B and 2B are small compared to commonly used models like 7B, 13B etc. It’s hard to conclude yet whether this strategy will scale well. So some scaling study will show whether this work can be really a stronger candidate to replace the baselines.

**Questions:**

1) What is the reason to choose UL2 objective for training with 60% causal LM, 20% prefix LM and 20% span corruption? Do you see similar phenomena by just training with causal LM? Authors are requested to provide some ablation study and discussion about this.
2) From Table 1 we see memorization performance of MIDAS drops compared to baseline. Does the authors have any intuition why this is happening?
3) What is the performance of GRADSTACK for 2B parameters model in Table 1?

**Limitations:**

This work shows limited improvement on memorization-based tasks.

---

> ### Author Rebuttal · Authors · 2024-08-07
>
> Thank you for the feedback and many suggestions. We tried to address them below by running new experiments.
>
> **Q1**: *The benchmarked models of size 1B and 2B are small compared to commonly used models like 7B, 13B etc. It’s hard to conclude yet whether this strategy will scale well.*
>
> A: Based on the suggestion, we pretrained an 8B model on 100B tokens (instead of 400B tokens in the interest of time) for baseline and MIDAS. We try three learning rates to observe the effect of hyperparameters and report the results below. We include the evaluations corresponding to the categories in Table 1. We also include a new category for story completion tasks (average of LAMBADA, Story Cloze, Hellaswag) that also require reasoning from context, inspired by other reviewer suggestions. The average column now refers to an average of 18 tasks instead of 15. Overall the trends are very similar to 1B and 2B models: at 1.24x speedup MIDAS has much better task average. Interestingly, MIDAS seems a lot more robust to the choice of learning rate. We will include these 8B experiments and a longer one in the revision.
>
> | Model | Loss (val) | Closed book QA | Open book QA | Math Word Problems | Story Completion | Task Average (18) |
> |--|--|--|--|--|--|--|
> | Baseline (LR=1e2) | 1.962 | 13.7 | 35.1 | 27.7 | 49.7 | 30.3 |
> | MIDAS (1.26x) (LR=1e2) | 1.917 | 16.1 | 39.1 | 36.0 | 54.2 | 35.5 |
> |--|--|--|--|--|--|--|--|
> | Baseline (LR=3e3) | 1.911 | 16.5 | 37.2 | 30.8 | 53.2 | 33.1 |
> | MIDAS (1.26x) (LR=3e3) | 1.905 | 18.2 | 39.2 | 30.9 | 54.7 | 34.3 |
> |--|--|--|--|--|--|--|--|
> | Baseline (LR=1e3) | 1.898 | 17.9 | 38.6 | 27.6 | 51.8 | 32.5 |
> | MIDAS (1.26x) (LR=1e3) | 1.909 | 17.3 | 36.3 | 33.0 | 55.8 | 34.2 |
>
>
> ---
>
> **Q2**: *What is the reason to choose UL2 objective for training with 60% causal LM, 20% prefix LM and 20% span corruption? Do you see similar phenomena by just training with causal LM?*
>
> A: The observations that MIDAS is better than gradual stacking, and the inductive bias of MIDAS towards improved reasoning, also hold for GPT-style causal language modeling. We started with the UL2 objective and stuck with it, partly because it also uses causal LM for 60% of the data. Based on the reviewer's suggestion, we ran causal LM training for the 1B model and reported the results below. We include the evaluations corresponding to the categories in Table 1 for Causal LM training of the 1B model and also the new categories described in the response to **Q1**. The trends are very similar to the UL2 models.
>
> | Model | Loss (val) | Closed book QA | Open book QA | Math Word Problems | Story Completion | Task Average (18) | Primitives |
> |--|--|--|--|--|--|--|--|
> | 1B LM Baseline | 2.43 | 12.3 | 33.9 | 30.3 | 42.8 | 30.4 | 51.6 |
> | 1B LM GradStack (1.33x) | 2.43 | 10.3 | 32.1 | 22.3 | 42.3 | 26.5 | 39.9 |
> | 1B LM MIDAS (1.33x) | 2.44 | 11.4 | 36.6 | 29.1 | 50.2 | 31.3 | 58.7 |
>
>
> ---
>
> **Q3**: *memorization performance of MIDAS drops compared to baseline. Does the authors have any intuition why this is happening?*
>
> A: This is an interesting open question and we attempted an initial analysis in Section 4.2, where we notice a clear inductive bias of MIDAS towards improving open book QA problems (which is closer to reasoning) more than closed book versions of the same questions (which is closer to memorization). Our speculative hypothesis is that due to the connection to looped models, MIDAS has a slightly lower “effective” number of parameters, which is known to correlate with memorization abilities [1]. This seems to be more than compensated for by the improved reasoning abilities of MIDAS. We believe this phenomenon deserves further exploration and understanding.
>
>
> ---
>
> **Q4**: *What is the performance of GRADSTACK for 2B parameters model in Table 1?*
>
> A: Below we report results for GradStack with Prop-2 schedule for the 2B model, including the new task categories.
>
> | Model | Loss (val) | Closed book QA | Open book QA | Math Word Problems | Story Completion | Task Average (18) | Primitives |
> |--|--|--|--|--|--|--|--|
> | 2B Baseline | 1.926 | 15.2 | 39.1 | 27.1 | 54.4 | 32.4 | 54.4 |
> | 2B GradStack (1.24x) | 1.945 | 14.2 | 37.0 | 24.5 | 51.5 | 30.2 | 64.2 |
> | 2B MIDAS (1.24x) | 1.929 | 15.7 | 40.2 | 38.3 | 53.6 | 36.3 | 78.3 |
>
>
>
>
> [1] Allen-Zhu, Li. Physics of Language Models: Part 3.3, Knowledge Capacity Scaling Laws. 2024

---

> > ### Comment · Reviewer_s9u7 · 2024-08-13
> >
> > Thanks to the authors for answering my questions and validating them with further experiments. I think this is an interesting work which would drive further research, so I am keeping my score.

---

### Official Review · Reviewer_48NW · 2024-07-12

**Soundness:** 4
**Presentation:** 4
**Contribution:** 3
**Rating:** 7
**Confidence:** 3

**Summary:**

The authors propose MIDAS, an efficient and effective framework for gradually increasing model depth. Their method achieves better performance on some reasoning primitive problems. Also, the authors further provides empirical analysis to support their findings.

**Strengths:**

1. The authors propose MIDAS, a novel variant of gradual stacking, which achieves better training efficiency that baselines.
2. Their experiments show that MIDAS significantly outperforms baselines on certain reasoning primitive problems.

**Weaknesses:**

Can the authors evaluate on more reasoning and common-sense questions for better comparison, such as tasks tested in the Llama paper?

Minor:
In line 105, “For simplicity, we L is divisible by k” should be “For simplicity, L is divisible by k.”

**Questions:**

Will the authors release the code and dataset to allow the community to further study the interesting phenomenon mentioned in section 5?

Can this phenomenon be verified with fewer resources?

**Limitations:**

The authors need to add a separate section for limitations.

---

> ### Author Rebuttal · Authors · 2024-08-07
>
> Thank you for the feedback and suggestions. We tried to address them below by running new experiments.
>
> **Q1**: *Can the authors evaluate on more reasoning and common-sense questions for better comparison, such as tasks tested in the Llama paper?*
>
> A: Based on the reviewer’s suggestion, we run evaluations on common sense reasoning benchmarks PiQA, HellaSwag, Winogrande, ARC-E, ARC-C from the Llama paper.
>
> | Model | PiQA | HellaSwag | Winogrande | ARC-E | ARC-C | Average (5) |
> |--|--|--|--|--|--|--|
> | 1B Baseline | 75.4 |	58.7 |	59.8 |	63.2 | 	31.7 | 57.8 |
> | 1B MIDAS Prop-2 (1.33x) | 74.0 | 58.9 | 58.9 | 63.1 | 31.4 | 57.3 |
> | 1B MIDAS Prop-3 (1.25x) | 74.0 | 60.4 | 59.0 | 62.5 | 32.7 | 57.7 |
> |--|--|--|--|--|--|--|
> | 2B Baseline | 75.6 | 65.0 | 62.3 | 66.9 | 35.2 | 61.0 |
> | 2B MIDAS Prop-2 (1.24x) | 76.0 | 65.5 | 62.8 | 67.0 | 34.5 | 61.1 |
>
> We did not include BoolQ because the evals are very noisy and close to “trivial” accuracy. Furthermore the trends are highly sensitive to which metric is used (accuracy vs auc-pr).
>
> Overall, we find the MIDAS is roughly neutral related to baseline at ~25% speedup. We do not observe a strong inductive bias for these tasks. Our hypothesis is that common-sense questions require a significant component of “memorization” of world knowledge of what is and what is not plausible in the physical world.
>
> ---
>
> **Q2**: *Will the authors release the code and dataset to allow the community to further study the interesting phenomenon mentioned in section 5?*
>
> A: We will release the reasoning primitives dataset and code to aid future research on this topic.
>
> ---
>
> **Q3**: *Can this phenomenon be verified with fewer resources?*
>
> A: The observation that MIDAS (middle stacking) is better than gradual stacking also holds at the 120M parameter scale, with decoder-only models like GPT-2 small and also with masked LM with BERT-Base. We did not report these results in the paper since the focus was on larger scale models, but we can include these results in the revision. The inductive bias towards reasoning and non-trivial performance on reasoning primitives may require closer to a billion parameters to see non-trivial few-shot performance.
>
> We will fix the typos pointed out by the reviewer and will include more discussion about limitations, particularly regarding the understanding of the inductive bias.

---

> > ### Comment · Reviewer_48NW · 2024-08-07
> > **Thanks for the rebuttal!**
> >
> > Thank you for the rebuttal! I find the paper interesting, although it is still in its early stages. I keep my score.

---

> ### Author Response · Authors · 2024-08-08
> **Response**
>
> Thank you for acknowledging our rebuttal and your continued interest! One point we forgot to highlight in our earlier response was the following: at **1.25x speed up**, MIDAS not only improves reasoning primitives, but also **significantly improves standard benchmarks** like **open book QA** tasks (includes TydiQA, SquadV2, DROP, QuAC, CoQA) , **story completion** (includes Lambada, StoryCloze, HellaSwag), **math word problem datasets** (ASDiv, MAWPS, SVAMP) and GSM8k finetuning, and is roughly neutral on closed book QA and commonsense tasks (based on results in Table 1). These speedup and quality improvements are also **verified at 1B, 2B and 8B parameter** scales. Our reasoning primitives were designed specifically to isolate the factors leading to reasoning benefits, and so the improvements there are even higher there.
>
> Hopefully this convinces the reviewer that the results are fairly mature not early stage.

---

### Official Review · Reviewer_1oxX · 2024-07-13

**Soundness:** 3
**Presentation:** 4
**Contribution:** 3
**Rating:** 6
**Confidence:** 4

**Summary:**

Gradual stacking involves incrementally growing a model by stacking its last few layers to initialize the next stage. A new variant called MIDAS (MIDdle grAdual Stacking) is proposed, which stacks the middle block of layers instead of the last block. This method is found to be more efficient and shows a bias towards improving reasoning tasks. MIDAS demonstrates an inductive bias that enhances performance on downstream tasks requiring reasoning, despite similar or slightly worse perplexity compared to baseline training. This inductive bias was analyzed using reasoning primitives (simple synthetic tasks) which revealed that models pretrained with stacking perform better on these primitives without fine-tuning.

**Strengths:**

1. This paper proposed a novel stacking algorithm for improving reasoning.
2. Experiments show that the algorithm improves performance on four distinct types of reasoning benchmarks.
3. The Deep dive into reasoning section shows very interesting insights on reasoning.

**Weaknesses:**

The authors showed empirical evidence that the proposed algorithm improves reasoning, but reduces memorization. However, the experiments are done on four distinct categories of reasoning benchmarks. I think at least one more reasoning benchmark is needed in each category in order to justify that the improvement difference is indeed due to memorization instead of some dataset artiacts.

**Questions:**

Does it make sense to test on existing synthetic reasoning benchmarks instead of just synthetic reasoning primitives? For example, ProofWriter: https://arxiv.org/abs/2012.13048.

**Limitations:**

Yes, the authors adequately addressed the limitations

---

> ### Author Rebuttal · Authors · 2024-08-07
>
> Thank you for the positive feedback and suggestions.
>
> **Q1**: *I think at least one more reasoning benchmark is needed in each category in order to justify that the improvement difference is indeed due to memorization instead of some dataset artifacts.*
>
> A: Based on the reviewer’s suggestion, we evaluated MIDAS and baseline models on another category of tasks called “story completion”, which includes tasks like LAMBADA, StoryCloze and Hellaswag. These measure the model's ability to correctly complete a story or premise. These tasks are also closer to reasoning, since the completion needs to be figured out from the given premise. The results for the 1B UL2 MIDAS models are presented below. Please refer to the response to Reviewer s9u7 for more evaluations on the story completion category.
>
> | Model | Lambada | StoryCloze | HellaSwag |
> |--|--|--|--|
> | 1B Baseline | 16.1 | 73.1 | 58.7 |
> | 1B MIDAS-Prop2 (1.33x) | 18.4 | 74.6 | 58.9 |
> | 1B MIDAS-Prop3 (1.25x) | 25.5 | 74.9 | 60.4 |
>
> **Q2**: *Does it make sense to test on existing synthetic reasoning benchmarks instead of just synthetic reasoning primitives? For example, ProofWriter*
>
> A: Thank you for the suggestion. One of the motivations behind new reasoning primitives was to evaluate these models on very simple and basic tasks that can isolate the benefits of MIDAS, which we could not do successfully with existing benchmarks. That said, it also makes sense to test on more complex synthetic tasks like ProofWriter. Unfortunately, we were not able to set up this evaluation task in time for the response since we prioritized some other experiments, but we will try to include this in the revision.

---

### Official Review · Reviewer_UVBV · 2024-07-21

**Soundness:** 3
**Presentation:** 3
**Contribution:** 3
**Rating:** 6
**Confidence:** 3

**Summary:**

The paper proposes an improvement of the gradual stacking method proposed in Reddi et al. 2023 for efficient training. The improved method relies on an observation that stacking the layers at the end exhibits the similarity between layers at the end and this might be a suboptimal choice but stacking the layers in the middle layers can exhibit the similarity between the layers in the middle layers (similar to looped models). Stacking the middle layers improves the performance on reasoning tasks such as open book QA and math word problem tasks. The paper also shows that even though the validation log perplexity of baseline and gradual stacking methods are similar to the proposed method, the proposed method achieves better performance on the downstream reasoning tasks. At the end, the paper proposes some simple tasks such as induction copying, variable assignment, and pre-school math tasks that might be responsible for improvement on reasoning tasks.

**Strengths:**

- The reasoning behind stacking in the middle rather than at the end is simple and seems effective in experiments.
- It is interesting that the paper tried to characterize the improvement of stacking in the middle by characterizing its connection to the looped transformers and showing the benefits in the reasoning-related tasks. The paper also provides extensive experiments to support their claims in this regard.
- The paper is well-written and easy to follow overall.

**Weaknesses:**

- In section 5, the paper conjectures that induction copying, variable assignment, and pre-school math tasks are some of the core capabilities of contextual reasoning that can help characterize the inductive bias of the proposed approach. The proposed method MIDAS achieves a significant improvement in these tasks. However, the improvement of MIDAS on downstream tasks (open book QA and math word problems) is marginal compared to the improvements on these tasks. This discrepancy raises the question if there are synthetic tasks for which MIDAS is not performing as well as the baseline which is affecting the performance on downstream tasks.

- There are some typos in the paper:
   - This on line 103?
   - Line 105?

**Questions:**

See the weakness section.

**Limitations:**

The paper discusses some of the limitations in reasonable detail.

---

> ### Author Rebuttal · Authors · 2024-08-07
>
> Thank you for the positive feedback and insightful questions.
>
> **Q1**: *the improvement of MIDAS on downstream tasks (open book QA and math word problems) is marginal compared to the improvements on these tasks [reasoning primitives].*
>
> A: While the improvement on these benchmarks is lower compared to improvement on primitives, the absolute magnitude of improvement is still quite high. For instance, for the 1B model, MIDAS with Prop-2 schedule improves open book QA from 33.3 -> 36.3 (+9% relative) and math word problems from 23.5 -> 29.0 (+23% relative). The reviewer is correct in observing that improvements on reasoning primitives from 35.2 -> 58.4 (+65% relative) are even larger. Our hypothesis is the following: solving each benchmark dataset requires some combination of different skills like memorization, reasoning, etc. with varying proportions. Based on the analysis from Section 4.2, we observed that MIDAS helps a lot more on tasks that require more reasoning (open book QA) than the memorization version of the same QA tasks. Reasoning primitives were designed to specifically isolate reasoning skills and minimize the need for memorization. Thus, we believe that MIDAS improves the most on primitives. Tasks like open book QA do require some reasoning from context, however, they can also benefit a lot from memorization of facts and world knowledge where MIDAS does not improve over baseline. (For instance the model may choose to ignore the context and answer from memory, since after all it has decent closed book QA accuracy). This, we believe, can dampen the magnitude of improvements. Math word problems intuitively require more reasoning than memorization compared to open book QA, and thus improvements are larger. We believe that understanding this interplay between reasoning and memorization is a very interesting and important future direction.
>
>
> **Q2**: *... raises the question if there are synthetic tasks for which MIDAS is not performing as well as the baseline*
>
> A: That is a good question. Since “memorization” is the slice where MIDAS has the least improvement (slight regression in some cases), we would need to construct a synthetic dataset that tests for memorization abilities. Even on closed book QA benchmarks, MIDAS is almost neutral in various cases, so finding a task where MIDAS is substantially worse would be an interesting direction. We could not come up with a very good synthetic task to isolate this effect, and if the reviewer has some suggestions, we are happy to try those.

---

> > ### Comment · Reviewer_UVBV · 2024-08-11
> >
> > Thank you for the rebuttal! It is good that similar results also hold when the model is trained with the Causal LM objective. I am keeping my current score.

---

### Author Rebuttal · Authors · 2024-08-07

We thank the reviewers for constructive feedback and their appreciation for the results in the paper. We have responded to reviewer questions independently. Following reviewer suggestions, we ran and reported the following new experiments:

- Run GradStack for the 2B UL2 model that was missing in Table 1. MIDAS continues to be much better than GradStack (see Q4 of reviewer s9u7)
- Pretraining with causal LM objective on 1B model. Trends are very similar to UL2 pretraining objective (see Q2 of reviewer s9u7)
- 8B pretraining with 100B tokens to verify that the results scale with model size. MIDAS @ 1.26x speedup has better downstream evals than baseline training, and is more robust to choice of learning rate. (see Q1 of reviewer s9u7)
- Add new evaluations on another category of tasks, “story completion”, that includes Lambada, StoryCloze and HellaSwag. The inductive bias of MIDAS towards reasoning shows up here too. (see Q1 of reviewer 1oxX)
- More evaluations on common sense tasks from Llama paper (PiQA, HellaSwag, WinoGrande, ARC-E, ARC-C). MIDAS @ 1.25x speedup is neutral on average compared to baseline training. (see Q1 of reviewer 48NW)


We hope this addresses any remaining concerns by the reviewers, and we are happy to engage in more discussions.

---

### Decision · Program_Chairs · 2024-09-25

**Decision:**

Accept (poster)

**Comment:**

A paper that contributes to the field.

The reviewers and myself were positive about the paper, recognizing its novel approach and its potential impact on improving training efficiency and reasoning capabilities in language models. The proposed MiDAS technique is seen as a significant improvement over previous stacking methods, particularly in enhancing performance on reasoning-related tasks, which is a critical area of research. The empirical results are solid, and the connection made between stacking and looped models is particularly insightful, paving the way for further research in this area. The paper is also well-written and easy to follow, which was noted by multiple reviewers.

Overall, strenghts:
Novelty and Effectiveness: The idea of stacking middle layers instead of the last few layers is both novel and effective, as demonstrated by the experiments. Empirical Validation: Extensive experiments validate the proposed method, showing significant improvements in reasoning tasks. Well-Written: The paper is clearly presented, making the complex concepts easier to understand.

The following are suggestions for minor expansions and clarifications:
The paper could benefit from additional evaluations on more diverse reasoning benchmarks to further substantiate the claim that MIDAS enhances reasoning over memorization. Consider adding synthetic reasoning benchmarks like ProofWriter to provide a broader perspective. Moreover, the drop in performance on memorization tasks, compared to baseline models, was noted. A more detailed analysis and possible mitigation strategies could strengthen the paper. Understanding and explaining this trade-off more deeply would be valuable.

In summary, this paper presents a well-supported and innovative approach to improving reasoning in language models and it will be a good contribution to the field.